# Genotype Characterization and MiRNA Expression Profiling in Usher Syndrome Cell Lines

**DOI:** 10.3390/ijms25189993

**Published:** 2024-09-17

**Authors:** Wesley A. Tom, Dinesh S. Chandel, Chao Jiang, Gary Krzyzanowski, Nirmalee Fernando, Appolinaire Olou, M. Rohan Fernando

**Affiliations:** Molecular Diagnostic Research Laboratory, Center for Sensory Neuroscience, Boys Town National Research Hospital, Omaha, NE 68010, USA; wesley.tom@boystown.org (W.A.T.); dinesh.chandel@boystown.org (D.S.C.); chao.jiang@boystown.org (C.J.); gary.krzyzanowski@boystown.org (G.K.); nirmalee.fernando@boystown.org (N.F.); appolinaire.olou@boystown.org (A.O.)

**Keywords:** Usher syndrome, biomarkers, miRNA, microarray, droplet digital PCR, exon sequencing

## Abstract

Usher syndrome (USH) is an inherited disorder characterized by sensorineural hearing loss (SNHL), retinitis pigmentosa (RP)-related vision loss, and vestibular dysfunction. USH presents itself as three distinct clinical types, 1, 2, and 3, with no biomarker for early detection. This study aimed to explore whether microRNA (miRNA) expression in USH cell lines is dysregulated compared to the miRNA expression pattern in a cell line derived from a healthy human subject. Lymphocytes from USH patients and healthy individuals were isolated and transformed into stable cell lines using Epstein–Barr virus (EBV). DNA from these cell lines was sequenced using a targeted panel to identify gene variants associated with USH types 1, 2, and 3. Microarray analysis was performed on RNA from both USH and control cell lines using NanoString miRNA microarray technology. Dysregulated miRNAs identified by the microarray were validated using droplet digital PCR technology. DNA sequencing revealed that two USH patients had USH type 1 with gene variants in USH1B (*MYO7A*) and USH1D (*CDH23*), while the other two patients were classified as USH type 2 (*USH2A*) and USH type 3 (*CLRN-1*), respectively. The NanoString miRNA microarray detected 92 differentially expressed miRNAs in USH cell lines compared to controls. Significantly altered miRNAs exhibited at least a twofold increase or decrease with a *p* value below 0.05. Among these miRNAs, 20 were specific to USH1, 14 to USH2, and 5 to USH3. Three miRNAs that are known as miRNA-183 family which are crucial for inner ear and retina development, have been significantly downregulated as compared to control cells. Subsequently, droplet digital PCR assays confirmed the dysregulation of the 12 most prominent miRNAs in USH cell lines. This study identifies several miRNA signatures in USH cell lines which may have potential utility in Usher syndrome identification.

## 1. Introduction

Usher syndrome (USH) is an autosomal recessive inherited disorder that profoundly impacts hearing, vision, and balance. It is characterized by the triad of sensorineural hearing loss (SNHL), vision impairment due to retinitis pigmentosa (RP), and vestibular dysfunction. The prevalence of USH ranges from 4 to 17 cases per 100,000 individuals [1,2,3]. USH is clinically categorized into three distinct types, distinguished by the age of onset of hearing and visual impairments, the severity of hearing loss, and the presence of balance issues. USH type 1 (USH1) is marked by early-onset SNHL, RP-related vision loss, and balance problems that commence within the first decade of life. USH type 2 (USH2) is characterized by the onset of mild to moderate non-progressive SNHL and RP between the ages of 10 and 20, accompanied by normal vestibular function. USH type 3 (USH3) involves progressive SNHL, sometimes accompanied by balance disturbances, with a variable onset time for SNHL and RP.

USH presents as a clinically and genetically diverse condition with nine confirmed causative genes. While traditionally considered a monogenic disorder, recent studies have unveiled instances of digenic inheritance [4,5]. Mutations in eight genes, including *MYO7A*, *USH1C*, *CDH23*, *PCDH23*, *USH1E*, *USH1G*, *USH1K*, and *CIB2*/*DFNB48*, are responsible for USH1 [1]. Of relevance to this study, Usher 1 subtypes (Usher-1D and Usher-1B) are indeed classified by the genes which possess the dominant pathogenic mutation, although there are digenic instances reported in Usher patients. Usher-1B (*MYO7A*) mutations make up 53–70% of Usher 1 pathogenic variants, while Usher-1D (*CDH23*) mutations account for 10–20% of pathogenic variants in Usher 1 patients [6]. USH2 results from genetic mutations in *USH2A*, *GPR98* (also known as *ADGRV1*), and *WHRN* (also known as *DFNB31*) [1], while USH3 is attributed to mutations in the *CLRN-1*, *HARS*, and *PDZD7* genes [1].

Usher-1B is thought to be caused by mutations in the *MYO7A* gene. *MYO7A* encodes myosin VIIA, an unconventional mechanochemical protein containing a motor domain, an actin-binding domain, a neck domain, and a short tail domain [7]. In inner ear hair cells (IHC), myosin VIIA is thought to aid in correct localization of mechano-transduction channels and ankle links in stereocilia [8]. Mutations in *MYO7A* disrupt the ability of IHC stereocilia to function properly, resulting in progressive hearing loss. In the retina, myosin VIIA aids in the transport, distribution, and localization of molecules essential for photoreceptor cell function and retinal pigment epithelial (RPE) cell function [9]. Myosin VIIA mutations have been shown to result in degeneration of photoreceptor cell and RPE cell function over time [10].

Mutations in *CDH23*, a gene encoding an atypical cadherin protein, are thought to be responsible for the Usher-1D phenotype [6]. Cadherin-related 23 protein is a transmembrane protein with a chain of 27 extracellular cadherin (EC) domains, a transmembrane domain, and a PDZ binding motif thought to aid in anchoring to an actin cytoskeleton, and the atypically long extracellular domains (ECs) are involved in cell–cell adhesion [11]. In the inner ear, *CDH23* is expressed in sensory hair cells and appears to play an important role in the differentiation of hair cells, specifically the development of the stereocilia [12]. In the retina, *CDH23* is found primarily in connecting cilium, the basal body complex, and ribbon synapses of photoreceptor cells, maintaining structural proximity for proper synapse function [13,14]. Mutations in *CDH23* have been shown to disrupt the development of stereocilia in cochlear hair cells and can also result in degradation of retinal photoreceptor cell function [11].

The functional role of the protein usherin (*USH2A*) remains elusive; however, it is speculated to have a role in maintaining efficient cell signaling in ciliated cells [15,16]. It has also been observed that disruption of *USH2A* leads to loss of photoreceptor cells, causing retinal degradation over time, and in cochlear hair cells the absence of usherin during development leads to hair cell malformation and hearing loss [17].

Clarin-1 is known to be a glycosylated transmembrane protein which has amino acid sequence homology with stargazin (*Cacng2*), suggesting a potential role in hair cell and photoreceptor cell synapse function [11,18]. Until recently, the effects of *CLRN1* mutations on the retina have largely remained undefined, but recently severe photoreceptor degradation has been described in patients with *CLRN1* variants [19].

MicroRNAs (miRNAs) are small, single-stranded, non-coding RNA molecules spanning 21 to 23 nucleotides in length that play pivotal roles in post-transcriptional gene expression regulation. These miRNA molecules are integral regulators of cellular homeostasis in both normal physiological and pathological processes [20]. Extensive research has underscored the significance of miRNAs in the visual, auditory, and vestibular systems [21,22,23,24,25]. Specifically, the microRNA-183 family, encompassing miRNA-183, miRNA-182, and miRNA-96, is highly conserved and indispensable for the development and maturation of sensory organs [26]. This family of miRNAs plays crucial roles in maintaining normal growth and population of hair cells and neurons within the inner ear [27,28]. These are also vital for the maturation of photoreceptor cells [29,30] because disruptions in the miRNA-183 family have been implicated in syndromic retinal degradation and substantial vestibular impairments [31].

As growing miRNA research continues to elucidate their important roles in inner ear development and hearing loss, differentially expressed miRNAs thus may have potential future applications in diagnostics and treatment to restore hearing function [28]. However, defining precise roles and targets of individual miRNAs involved in neurosensory pathways in the inner ear remains a major challenge. MiRNA studies on hearing loss have relied mostly on animal models due to the inaccessibility of the human inner ear. MiRNAs may also play an important role in retinal photoreceptor cell maturation and functionality [29,30]. Dysregulation of these miRNAs may cause impaired vision, and hence they have the potential to be used to develop treatment strategies and diagnostic assays. The approach of this study, however, aimed to examine miRNA expression patterns in lymphocyte cell lines derived from Usher syndrome patients, comparing them with those from a healthy individual without hearing, vision, or vestibular impairments. We hope the findings from this preliminary research will assist in the development of miRNA-based tools for future diagnostic applications for Usher syndrome.

## 2. Results

### 2.1. Targeted Exon Sequencing

The genotype of all cell lines was confirmed using targeted exon sequencing as summarized in Table 1.

The D3739 cell line has a homozygous likely pathogenic G>A missense mutation in the *CDH23* gene and was thus attributed to the USH1D genotype. This genotype is consistently associated with USH1, where the variant creates a novel splice acceptor site which results in an in-frame deletion of 51 base pairs, removing a calcium-binding motif of the protein [3,32,33,34,35,36,37,38,39,40,41,42]. The D3741 cell line contains a homozygous likely pathogenic G>T missense mutation in the *MYO7A* gene. Therefore, the D3741 cell line was categorized under the USH1B genotype. This variant has been detected both as a homozygous mutation and as a compound heterozygous mutation. The USH1B cell line possesses a likely pathogenic homozygous missense variant in the *MYO7A* gene c.2905G>T (p.Glu968Asp) which is predicted to cause splice site variation in the myosin VIIA gene [37,40,43,44,45,46,47,48,49,50,51]. In cell line D2880, a homozygous variant in the *CLRN1* gene was detected at c.528A>C (p.Tyr176X), and thus it was categorized as USH3A. This mutation causes a premature stop codon, truncating the clarin-1 protein [52,53,54,55]. Interestingly, the USH2A cell line has a likely compound heterozygous mutation phenotype with two variants in the *USH2A* gene. The first variant, *USH2A* c.4338_4339del (p.Cys1447fs), is a frameshift variant which is predicted to cause a premature stop codon [3,15,21,45,56,57,58,59,60,61,62]. The second heterozygous variant has yet to be functionally confirmed. It occurs in *USH2A* c.14787del (p.Glu4930fs) and is computationally predicted to cause premature truncation or nonsense-mediated decay in the production of usherin protein [45,56,61,63].

### 2.2. miRNA Microarray Analysis

Normalized microarray data were further analyzed using DESeq2 to identify miRNAs that were differentially expressed in USH cell lines compared to control cells. At least a twofold increase or decrease with a BH-adjusted *p* value < 0.05 was considered as significantly altered miRNA expression. Using this criterion, we found 92 differentially expressed miRNAs in USH cells. Of these, 20 were unique to USH1, 14 to USH2, and 5 to USH3; 2 were unique to both USH1 and USH2, 5 to USH1 and USH3, and 10 to USH2 and USH3. The remaining 36 were identified as common to all USH types (Figure 1 and Table 2).

PERMANOVA analysis of miRNA microarray data confirmed that miRNA expression profiles were significantly different between all cell lines with a BH-adjusted *p*-value of 0.001 (Figure 2).

Results obtained using PERMANOVA were corroborated by PCA analysis, showing distinct clustering by cell line. However, USH2A and USH3A clustered more similarly compared to controls or USH1 phenotypes, and a total of 94.8% of the variation in the miRNA expression profile dataset was explained by the ordination of the first two principal components (Figure 2). Additionally, of the top 6 miRNAs driving variation between samples, hsa-miR-155-5p, hsa-miR-142-3p, and hsa-let-7a-5p were differentially expressed among Usher cell lines, particularly in USH2A and USH3A samples. Variation in levels of hsa-miR-16-5p and hsa-miR-19b-3p was assigned to the control cell lines. The hsa-miR-4454+hsa-miR-7975 was different in one of the USH2A sample replicates.

Table 3 provides a list of the top 12 differentially expressed miRNAs in USH cells compared to controls.

The heatmap (Figure 3) shows differential expression of these miRNAs uniquely assigned to USH and control cell lines.

All miRNAs in this list were differentially expressed in the DESeq2 analysis, with a BH-adjusted *p*-value < 0.05, and had at least a twofold change in miRNA expression between Usher cell lines and controls. Additionally, these miRNAs had an average copy number greater than 50 within the dominant sample type. The following six miRNAs, hsa-miR-16-5p, hsa-miR-19b-3p, hsa-miR-4454+hsa-miR-7975, hsa-miR-142-3p, hsa-miR-155-5p, and hsa-let-7a-5p, were also the top six miRNAs explaining high amounts of variation in miRNA profiles in the PCA biplot (Figure 2). Both PCA and the heatmap showed USH2 and USH3 grouped together. Control samples clustered together, and USH1B and USH1D samples shared similar miRNA expression profiles (Figure 3). The top 12 miRNAs panel examined includes 6 downregulated and 6 upregulated miRNAs. In addition, miRNA-28-5p, miRNA-96-5p, miRNA-182-3p, miRNA-183-3p, miRNA-16-5p, and miRNA-19b-3p were downregulated in all USH types compared to controls (Figure 4).

According to Figure 5A,D,F, miRNA-363-3p, miRNA-155-5p, and miRNA-142-3p were upregulated in all USH types. MiRNA-223-3p (Figure 5B) was upregulated only in USH2A and USH3A, whereas miRNA-150-5p showed increased expression only in USH2A (Figure 5C). MiRNA-let7a-5p is upregulated in USH2A, USH3A, and USH1B (Figure 5E). However, this change was not evident in USH1D (Figure 5E), showing that same phenotypes with different genotypes may vary in their miRNA expression patterns.

### 2.3. miRNA ddPCR Analysis

The top 12 differentially expressed miRNAs identified after microarray screenings were validated using droplet digital PCR technology. Figure 6 shows the expression of six miRNAs that were significantly downregulated in USH cells compared to controls.

These miRNAs are miRNA-28-5p, miRNA-96-5p, miRNA-182-3p, miRNA-183-3p, miRNA-16-5-p, and miRNA-19b-3p. The first four miRNAs—miRNA-28-5p, miRNA-96-5p, miRNA-182-3p, and miRNA-183-3p—showed no statistically significant difference among USH types (Figure 6A–D). However, miRNA-16-5-p and miRNA-19b-3p showed variation within USH types: USH1D showed relatively more downregulation than USH2A (Figure 6E,F). Figure 7 shows the expression of six miRNAs that were upregulated in USH cells compared to controls.

Expression levels of miRNA-363-3p were significantly higher in all USH types except for USH3A (Figure 7A). Similarly, miRNA-223-3p was significantly upregulated in all USH types except for USH1B (Figure 7B). Statistically significant upregulation of miRNA-150-5p was observed only in USH2A (Figure 7C). MiRNA-155-5p, miRNA-let-7a-5p, and miRNA-142-3p were upregulated in all USH types compared to the control (Figure 7D–F).

## 3. Discussion

In this study, we assessed the expression profiles of 798 miRNAs across different Usher and control cell line types, revealing intriguing variations. The Usher cell lines exhibited distinct miRNA expression profiles, as illustrated in Table 2. These variations were not only evident among Usher type 1, 2, and 3 phenotypes but were also discernible between the USH1B and USH1D genotypes. According to microarray data (Figure 2), the miRNA expressions in all Usher samples significantly differed from those in the control samples. We hypothesize that these differences in miRNA expression may be further complicated by unique combinations of sequence variants contributing to the phenotype. For instance, in the USH2A cell line, we identified two heterozygous frameshift mutations that appear to have a compounded effect, resulting in the observed phenotype. This compound mutation effect, although somewhat unconventional, has been documented in Usher syndrome, challenging the widely accepted notion that homozygous recessive mutations exclusively underlie the disorder [33,64,65]. In contrast, USH1D, USH1B, and USH3A all exhibited homozygous recessive mutations, aligning with the classical definition of autosomal recessive inheritance in Usher patients [34,44,65].

The miRNA-183/182/96 family emerged as a key player, showing significant downregulation in all Usher cell lines compared to the control, as is evident from both DESeq2 analysis of microarray results (Figure 4B–D) and ddPCR analysis (Figure 6B–D). Previous studies have showcased the significance of these miRNAs, as their inactivation in mice led to significant developmental issues in cochlear hair cells [66]. Moreover, mutations affecting miRNA-96-5p have been linked to impaired growth and maturation and, in some cases, malformation of cochlear hair cells in mice and humans [67,68,69,70]. Additionally, alterations in the expression of the microRNA-183/182/96 family have been associated with retinal dystrophy—also a component of Usher syndrome [30,31,71]. Our findings align with the existing literature, supporting the observation that microRNA-183/182/96 downregulation is consistent across various Usher genotypes and thus may serve as an indicator for Usher syndrome. Notably, our microarray data revealed an average 2.43 ± 0.22 log2-fold decrease in miRNA-183-3p expression, a 2.98 ± 0.27 log2-fold decrease in miRNA-96-5p expression, and approximately a 6.83 ±1.05 log2-fold decrease in miRNA-182-5p expression in Usher samples compared to controls (Table 3). This downregulation across the entire miR-183/182/96 family was confirmed using ddPCR, where Tukey’s HSD post hoc tests indicated significantly higher expression of all three miRNAs in control samples compared to Usher lines, with no significant differences observed among Usher lines (Figure 6B–D).

Interestingly, we observed a significant downregulation of miRNA-28-5p in all Usher samples, with an average 2.22 ± 0.31 log2-fold lower expression compared to controls (Table 3). Although no prior associations between miRNA-28-5p and Usher syndrome have been reported, Ji et al. (2017) suggested that miRNA-28 may target and regulate the expression of the cone-rod homeobox gene (CRX), making it a potential candidate for retinal degeneration, a component of Usher syndrome [72].

MiRNA-16-5p emerged as a driver of variation, particularly for control samples, displaying an average 1.51 ± 0.60 log2-fold higher expression in control samples compared to Usher lines (Table 3). Interestingly, some literature links elevated miRNA-16-5p expression to noise-induced hearing loss (NIHL) and even Alzheimer’s disease [73,74,75]. However, our study indicated low levels of miRNA-16-5p in Usher samples compared to controls, suggesting that the hearing loss caused by Usher and NIHL may have different mechanisms.

Six miRNAs, namely, miRNA-150-5p, miRNA-155-5p, miRNA-363-3p, miRNA-223-3p, miRNA-let-7a-5p, and miRNA-142-3p, exhibited upregulation in the Usher cell lines compared to the control (Table 3). Among these, miRNA-155-5p, miRNA-142-3p, and miRNA-let-7a-5p were the most abundant miRNAs in the dataset. Notably, miRNA-142-3p made up approximately 37.3% of miRNA copies in Usher samples, compared to 18.5% in the control line (Appendix A). This miRNA displayed an average 1.26 log2-fold higher expression in Usher samples as compared to the control. While miRNA-142-3p has not been previously associated with Usher or sensorineural hearing loss (SNHL), its downregulation has been observed in the aqueous humor of patients with central retinal vein occlusion (CRVO), a condition occasionally linked to retinitis pigmentosa (RP) [76,77]. Interestingly, our study found miRNA-142-3p to be one of the most abundant miRNAs in all samples, particularly in Usher cell lines (~40% of all miRNA copies), which contrasts with observations of CRVO (Appendix A).

Additionally, miRNA-let-7a emerged as a prominent driver of variation in both PCA and DESeq2 analyses, exhibiting an average 1.6 ± 0.61 log2-fold increase compared to control samples (Table 3). In Usher samples, miRNA-let-7a accounted for 2.35% of the total miRNA copies per genotype, while control samples displayed only a 0.9% proportion. Analysis of the heatmap revealed that both USH2A and USH3A exhibited the highest levels of miRNA-let-7a and miRNA-142-3p, while USH1 genotypes displayed lower expression (Figure 3), especially USH1D. This observation is consistent with the trajectory of miR-let-7a and miR-142-3p vectors in the PCA biplot (Figure 2).

In addition to examining miRNA expression patterns, we conducted a pathway analysis to summarize which metabolic pathways might be influenced by the target panel miRNAs (Appendix A). While there appears to be a diverse array of possible pathways the miRNA panel from this study could be affecting, we would like to consider two limitations. First, the samples from the study are immortalized B-lymphocytes which are not taken from the inner ear or retinal cells most affected in Usher syndrome. Thus, the pathway analysis provided in the supplementary material should be interpreted with caution due to the nature of the cell lines. Secondly, it should be emphasized here that this study is not aimed at identifying the specific function of the miRNAs we identified and their gene targets, but rather whether the miRNAs in this study can consistently differentiate between Usher and control cell lines. Although not necessarily responsible for the inner ear and/or retinal pathology, the identified miRNAs dysregulated in B-lymphocytes cell lines may have potential utility in the identification of Usher syndrome.

Targeted exome sequencing identified four distinct genotypes and likely pathogenic sequence variants in cell lines derived from clinically diagnosed Usher syndrome patients: *MY07A*, *CDH23*, *USH2A*, and *CLRN1* (Table 1). These genes encode proteins that are critical for the development and/or maintenance of inner ear hair cells and retinal photoreceptors [7,8,9,10,11,12,13,14,15,16,17,18,19]. However, the extent to which each patient from these cell lines experienced RP or progressive hearing loss is unknown. Therefore, further exploration of miRNA expression across a larger cohort of patient genotypes is essential to validate the top miRNA trends identified in this study. Nonetheless, our microarray results aligned with ddPCR quantification of the top 12 miRNAs. These 12 miRNAs, characterized by their consistency across both statistical tests and platforms (ddPCR and microarray), are significantly altered in four unique genotypes representing USH1B, USH1D, USH2A, and USH3A subtypes. Some miRNAs, such as the miR-183-182-96 family, are downregulated in all Usher types, while others, like miR-223-3p, exhibited subtype-specific regulation—being highly upregulated in the USH2A genotype but not in the other lines (Figure 4, Figure 5, Figure 6 and Figure 7).

It is important to acknowledge certain limitations in our experiments. Firstly, miRNA expression may exhibit tissue specificity. Data exclusively collected from immortalized lymphocyte cell lines may not necessarily reflect miRNA expression in other tissues within the same patient or in non-immortalized cells [78]. In addition, EBV transformation of lymphocytes has been reported to cause changes in miRNA levels. Specifically, miRNA-155, also known to be affected due to EBV, is thought to play a role in immune evasion and cell proliferation [79,80,81,82]. Interestingly, miRNA-155 is a member of our panel, yet its expression in the control cell line (NBT) remains highly downregulated. So, despite what is reported in the literature, even within our immortalized cell types there is still an expression discrepancy between the Usher and control lines, even in miRNAs known to be changed due to EBV infection. Because we knew that EBV causes changes in miRNA expression profiles in lymphocytes, we used normal immortalized cells as our control. To establish a robust control, we obtained B-lymphocytes from a healthy donor (age 43) with no history of hearing loss, vestibular dysfunction, or visual impairment. These cells were immortalized using Epstein–Barr Virus (EBV), creating a ‘normalized’ healthy control line. This method mirrors the same immortalization technique applied to generate the other cell lines used in this study. Importantly, B-lymphocytes are known to release microRNAs (miRNAs), which play a crucial role in gene regulation, including in the development and function of these cells [83,84]. Using a similarly immortalized control ensures that differences in miRNA expression between the cell lines are attributed to the disease state rather than the immortalization process itself. Both normal hearing control samples as well as Usher patient-derived lymphocytes were purified from blood and immortalized following standard EBV-transformation protocol. Therefore, we suspect that all miRNAs shown to express differentially in Usher cells compared to controls do so due to inherent pathologies or gene mutations related to Usher syndrome.

MicroRNA expression patterns may likely vary depending on the patient’s genotype. In our study, we observed significant differences in miRNA profiles within both the USH1B and USH1D genotypes, despite both falling under the USH1 classification. This divergence is attributed to distinct sequence variants found in different genes, contributing to the observed phenotypic differences. Here we document only four of the many different genotypes which may cause Usher syndrome, acknowledging there is much more work to be done in associating all genotypes and miRNAs. For example, in Figure 3, miRNA-150-5p would be an indicator of the USH2A genotype, while miRNA-363-3p serves as a better indicator of the Usher type 1 genotypes (*MYO7A* and *CDH23*). However, one of the main observations from this study is that there appear to be generalizations we can make about the miRNAs selected. First, looking at cluster A in Figure 3, there are six miRNAs (28-5p, 183-5p, 182-5p, 96-5p, 16-5p, and 19b-3p) which are significantly downregulated in all Usher genotypes. Thus, we might be able to use this set of six downregulated miRNAs to diagnose a potential Usher patient from a blood draw at an early age. If we examine block B in Figure 3, miRNAs 155-5p, 142-3p, and let-7a-5p all are significantly upregulated in all Usher patients compared to our control. There are generalized miRNA expression patterns that are consistent among multiple genotypes which can potentially be used as broad indicators of Usher syndrome. The top 12 differentially expressed miRNAs demonstrated robust performance across multiple statistical tests, with some having prior support documented in the literature. Thus, these miRNAs warrant further investigation as to their expression in Usher patient samples and across more Usher genotypes to assess their validity as indicators of disease.

## 4. Materials and Methods

### 4.1. Cell Lines

Three of the cell lines utilized in this study, namely, D3741 (USH1), D3739 (USH1), and D2880 (USH3), were originally established by Dr. William J. Kimberling’s laboratory at Boys Town National Research Hospital in Omaha, NE, USA. These cell lines were created through the infection of lymphocytes obtained from individuals with Usher syndrome (USH) using Epstein–Barr virus (EBV) derived from the B95-8 cell line. Informed consent was obtained from all donors prior to blood drawing, and the study was approved by the Institutional Review Board at Boys Town National Research Hospital (IRB#96-06-0X). For these three patients, we do not have the age at diagnosis. Blood was drawn from these patients just after diagnosing them as Usher patients. As part of our control group, lymphocytes were also isolated from a 43-year-old healthy donor and subsequently immortalized using EBV. In addition, a lymphocyte cell line corresponding to Usher syndrome type 2A (USH2A) was procured from the Coriell Institute (Catalog ID: GM09053, Camden, NJ, USA). This cell line was established from a 9-year-old patient. These cell lines were cultivated in RPMI 1640 medium, supplemented with 20% fetal bovine serum (FBS) and 50 µg/mL of gentamicin. Cultures were maintained in 100 × 20 mm tissue culture plates in a humidified atmosphere containing 5% CO_2_, at 37 °C.

### 4.2. DNA Extraction and Shearing

Genomic DNA was extracted from cultured cells by employing the QIAamp DNA Mini kit (Cat. No. 56304), following the manufacturer’s recommended protocol. Purified DNA was quantified using the “Qubit dsDNA BR Assay Kit” on a Qubit 4.0 fluorometer. To attain an optimal fragment size conducive to downstream applications, the genomic DNA was sheared using a Covaris M220 sonicator (Covaris LLC., Woburn, MA). Shearing resulted in fragments of approximately 250 bp. The effectiveness of the shearing process was confirmed through fragment size analysis using a D1000 Tapestation (P/N: 5067-5583, 5067-5582, Agilent Technologies). For each specific cell line under investigation (USH1B, USH1D, USH2A, and USH3A), 200 ng of sheared genomic DNA prepared in a final volume of 50.0 µL 0.1X-TE buffer was used for targeted exon sequencing.

### 4.3. Targeted Exon Sequencing

A custom exon probe panel targeting the coding regions of 26 genes associated with syndromic and non-syndromic hearing loss was created using Agilent’s SureDesign probe design tool (Agilent Technologies, Santa Clara, CA, USA). In total, 5740 biotinylated mRNA probes were synthesized, covering the coding regions of 26 genes of interest (200.345 Kbp total length) and used in a hybridization and capture approach for enrichment of target DNA. Probes covered exonic regions including a 25bp extension at 3′ and 5′ UTRs. Coding regions with >40 bp gaps in coverage received custom boosting and tiling strategies to ensure sufficient sequencing coverage. Genes covered in the panel include: *ABHD12*, *ADGRV1*, *ARSG*, *CDH23*, *CIB2*, *CLRN1*, *ESPN*, *FOXI1*, *GJB2*, *GJB6*, *HARS1*, *KCNE1*, *KCNJ10*, *KCNQ1*, *MYO7A*, *PCDH15*, *PDZD7*, *SLC26A4*, *USH1C*, *USH1G*, *USH2A*, *WHRN*, *CEP250*, and *CEP78*.

Libraries for the hearing loss exon capture were prepared using Agilent’s SureSelect XT HS2 DNA with a post-capture pooling protocol, as per the manufacturer’s instructions (P/N: G9985D, Agilent Technologies, Santa Clara, CA, USA). Briefly, 200 ng of extracted DNA in a total volume of 50 µL 0.1X TE buffer, sheared to ~250 bp with a Covaris M220 sonicator, was used as the starting input for library preparation (Covaris LLC, Woburn, MA, USA). Fragmented DNA samples underwent enzymatic end-repair followed by a dA-tail ligation. Sequencing adapters were ligated to dA-overhang, and each cell line received a unique dual-indexed primer pair with unique molecular indices. An 8-cycle PCR amplification of adapter ligated libraries was performed under the following conditions: 1 cycle: 98 °C (2 min); 8 cycles: 98 °C (30 s), 60 °C (30 s), and 72 °C (1 min); and 1 cycle: 72 °C (5 min). Indexed libraries were then hybridized with a biotinylated custom HL probe panel, and DNA-probe hybrids were captured using streptavidin beads (P/N: 65601, Thermo Fisher Scientific, Pittsburgh, PA, USA). Targeted libraries were purified using AMPure XP (P/N: A63880, Beckman Coulter Genomics, Palatine, IL, USA) 1X bead clean up, and library quality was assessed using a D1000 High Sensitivity Tapestation assay (P/N: 5067-5585, 5067-5584, Agilent Technologies, Wilmington, DE, USA). Post-capture libraries were diluted to equimolar concentrations and pooled for sequencing on an Illumina NextSeq 550DX 300-cycle high output flow-cell, with 150 bp paired end reads (P/N: 20024908, Illumina inc., Chicago, IL, USA). Targeted exon sequencing resulted in samples—USH3A (695.41 Mbp), USH1B (712.23 Mbp), USH1D (796.93 Mbp), and USH2A (767.00 Mbp)—with an average of 20.04 million reads per sample.

### 4.4. Variant Calling and Interpretation

Reads for each cell line were aligned and mapped to the human reference genome (GRCh38p.14) [85] using BWA and Samtools, respectively [86,87]. Sequence variants were called using Sentieon’s DNA pipeline for variant detection [88]. Sequencing variant interpretation according to ACMG classification criteria was conducted using the VarSeq software version 2.4.0 (Golden Helix enabling precision medicine, Available online: https://www.goldenhelix.com, accessed on 23 December 2023) [32,89].

### 4.5. RNA Extraction and miRNA Expression Assay

Briefly, total RNA was extracted from cell lines using QIAzol^®^ reagent (cat. # 79306), followed by purification using the miRNeasy Tissue/Cells Advanced Micro Kit (cat. #217684) protocol as per the manufacturer’s instructions (QIAGEN Sciences Inc., Germantown, MD, USA). Purified total miRNA from cell lines (100 ng miRNA per sample) was used as input for miRNA expression analysis. MiRNA expression in Usher and control cell lines was quantified using the NanoString© Human v3 miRNA assay (cat. # CSO-MIR3-12), performed on the nCounter Pro analysis system (NanoString Technology, Seattle, WA, USA). The assay detects 798 known human miRNAs, where each miRNA has specific tag sequences ligated with fluorescently barcoded reporter probes. After a hybridization period of 16 h at 65 °C, these miRNA-specific barcodes were detected by the nCounter Digital Analyzer, providing miRNA copy numbers. Raw count data from the miRNA assay were normalized using the NanoString quality control dashboard (NACHO version 2.0.0) package in R [90,91]. There are no well-established “housekeeping” miRNAs, so NACHO’s housekeeping predict = TRUE function was used to select the top five housekeeping miRNA candidates directly from the NanoString assay. The miRNA count data were normalized relative to internal positive, negative, and housekeeping predictions using a geometric mean normalization method = “GEO” [91]. The resulting normalized count table was used for downstream analysis in R.

### 4.6. Absolute Quantification of miRNAs by Droplet Digital PCR (ddPCR)

Following miRNA panel screening by the NanoString microarray, 12 miRNAs (6 upregulated and 6 downregulated) with significant differential expression patterns were selected for ddPCR analysis. For ddPCR, an equal starting RNA amount (10 ng) from each sample was first converted into cDNA by reverse transcription (RT) using miRNA-specific primer sets (Life Technology Corp., Carlsbad, CA, USA), following the TaqMan^TM^ RT kit protocol (Applied Biosystems Cat.#4366596). The cDNA/RT reaction was carried out in a total volume of 15 µL consisting of 2 µL of RNA template (5 ng/µL), 3 µL of RT-specific miRNA primers (5×), 0.15 µL dNTP-mix (25 mM each), 0.19 µL of RNAse-H, 1 µL of Multiscribe^TM^ reverse transcriptase enzyme (50 U/µL), and 1.5 µL of RT-buffer (10×), with the final reaction volume adjusted with nuclease-free water (7.16 µL). RT reactions were performed on a PCR cycler (cfx1000, BioRad, Richmond, CA, USA) with a set parameter of 16 °C for 30 m, 42 °C for 30 m, 85 °C for 5 m, and an infinite hold at 4 °C. The resulting cDNA products were diluted based on relative abundance of individual miRNAs. The standard ddPCR reaction was performed in a 20 µL reaction volume by adding 10 µL of Bio-Rad 2× ddPCR Supermix for probes, 2.0 µL of diluted cDNA, and 1.0 µL of individual miRNA-specific primer–probe mix, with the reaction volume adjusted to 20 µL using nuclease-free water. Droplet digital PCR was performed using a Bio-Rad Automated QX200 droplet digital PCR system as previously described [92]. MicroRNA copy numbers per 1 ng RNA were calculated using the equation given below:$$\begin{array}{cc} \;\mathrm{Copies}\;\mathrm{per}\;ng\;RNA & =\frac{\left(\left(\frac{ddPCR\;\mathrm{copy}\;\mathrm{number}\;}{ddPCR\;\mathrm{sample}\;\mathrm{volume}\;(\mu l)}\right)\;\times \;c\mathrm{DNA}\;\mathrm{dilution}\;\mathrm{factor}\;\times \;c\mathrm{DNA}\;\mathrm{stock}\;\mathrm{volume}\;(\mu l)\right)}{RNA\;\mathrm{input}\;(ng)} \end{array}$$

### 4.7. Statistical Analysis

Normalized count data were imported into the R statistical analysis software Version 4.2.2, where all subsequent microarray analysis was performed [90]. Usher-1D and Usher-1B cell lines had three replicates in the NanoString assay. Usher-2A, Usher-3A, and the control cell lines had four replicates each.

Differentially expressed (DE) miRNA analysis was performed using the “DESeq2” R package [19]. Venn diagrams depicting overlapping and unique miRNAs from DESeq2 differential analysis were produced using the ggvenn package [93]. All pairwise contrasts of phenotype (Usher-1B, Usher-1D, Usher-2A, Usher-3A, and NBT) were considered for differential miRNA expression analysis, and significance was assigned at a Benjamini–Hochberg (BH)-corrected *p*-value < 0.05.

Heatmaps of top differential miRNAs from pairwise genotype and phenotype contrasts were created using the “ComplexHeatmap” package in R. For heatmaps, expression count data were scaled for large differences in variation using the formula z_i_ = (x_i_ − u_i_)/s_i_, where x_i_ is the count for miRNA_i_, u_i_ is the mean for miRNA_i_, and s_i_ is the standard deviation of miRNA_i_ [94]. Samples were grouped using hierarchical clustering according to miRNA expression patterns [94].

Principal component analysis (PCA) was performed using a Bray–Curtis dissimilarity matrix calculated from normalized count data, and principal components were visualized with the “microViz” package version 0.12.5 [95]. Significant differences between miRNA profiles were calculated using a permutational multivariate analysis of variance (PERMANOVA) on Bray–Curtis dissimilarity distances with the adonis2 function in R’s “vegan” package [96,97]. The PERMANOVA used a generalized linear model considering the effect of phenotype (model formula: Bray–Curtis dissimilarity matrix ~ Phenotype).

For ddPCR analysis, a one-way ANOVA with a Tukey’s test for pairwise mean comparisons was used to determine differential expression between all cell lines, where statistical significance was assigned at *p* value < 0.05.

## 5. Conclusions

This cell-line-based study describes differential expression patterns of 12 miRNAs that may have important roles in the pathophysiology of Usher syndrome. Adopting both microarray and ddPCR techniques, we have shown expression levels of six upregulated and six downregulated miRNAs using transformed lymphocyte cell lines from Usher patients and healthy controls. While some of these miRNAs showed unique expression patterns in all Usher types compared to control, others were distinctly assigned between Usher subtypes (1B and 1D). However, our findings necessitate further investigations involving actual patient populations, which may help unravel the miRNAs’ regulatory mechanisms in neurodegenerative disorders impacting vision and hearing loss.

## Figures and Tables

**Figure 1 ijms-25-09993-f001:**
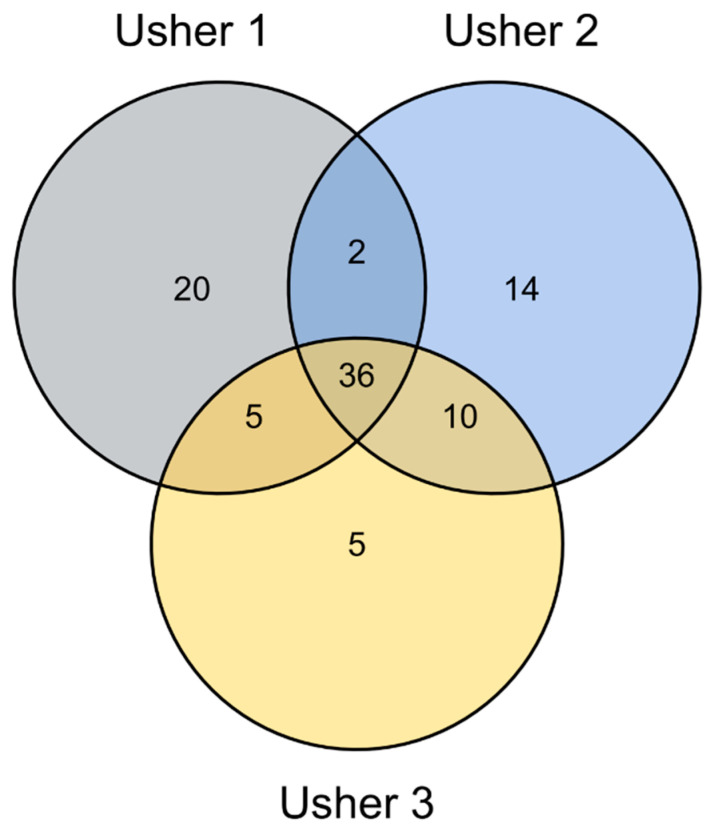
Venn diagram of miRNAs that are differently expressed in USH phenotypes. Venn diagram depicting the distribution of shared and unique differentially expressed miRNAs according to DESeq2 analysis. Among the 798 miRNAs detected by microarray, 92 miRNAs were differently expressed in USH phenotypes. Among these 92 miRNAs, 20 were specific to USH1, 14 to USH2, 5 to USH3, 2 to USH1 and USH2, 5 to USH1 and USH3, and 10 to USH2 and USH3; the remaining 36 were common to all USH types.

**Figure 2 ijms-25-09993-f002:**
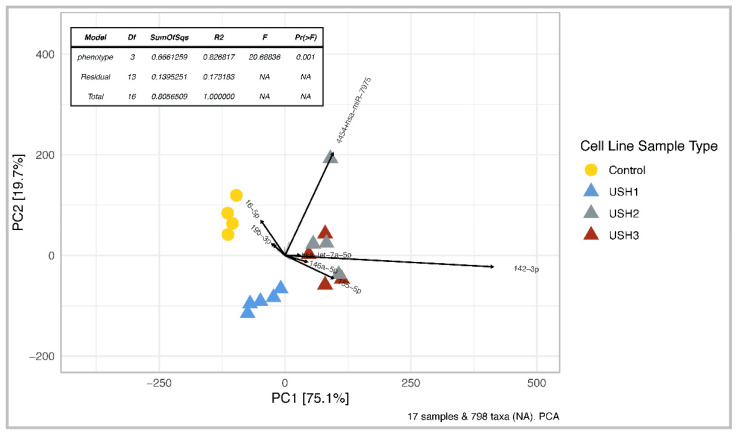
PCA biplot showing differences between miRNA expression profiles. Each point represents the miRNA profile of a sample; yellow circles are control cell lines; colored triangles represent Usher cell lines: USH1 (blue), USH2 (grey), USH3 (brown). Vectors labeled with miRNA are the top 6 miRNAs contributing to variation in the first two principal components. The table indicates the results from PERMANOVA analysis, where Pr (>F) indicates significance at values less than 0.05.

**Figure 3 ijms-25-09993-f003:**
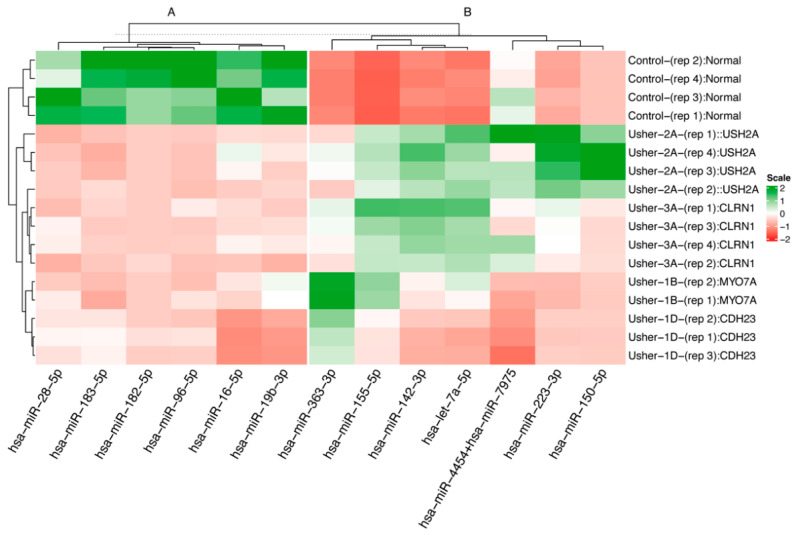
Heatmap of differential miRNAs of interest from DESeq2 contrasts of genotype and phenotype. Red tiling indicates downregulated miRNA expression, while green tiling indicates upregulated miRNA expression. Rows and columns were clustered by hierarchical clustering. Vertical clusters (**A**) and (**B**) show relationships between expression patterns, predominantly separating miRNAs upregulated in the control group in cluster (**A**), and upregulated miRNAs in the Usher phenotypes are represented in cluster (**B**). Row names in the heatmap denote the phenotype of the cell line and which replicate, as well as the genotype reported from exome sequencing.

**Figure 4 ijms-25-09993-f004:**
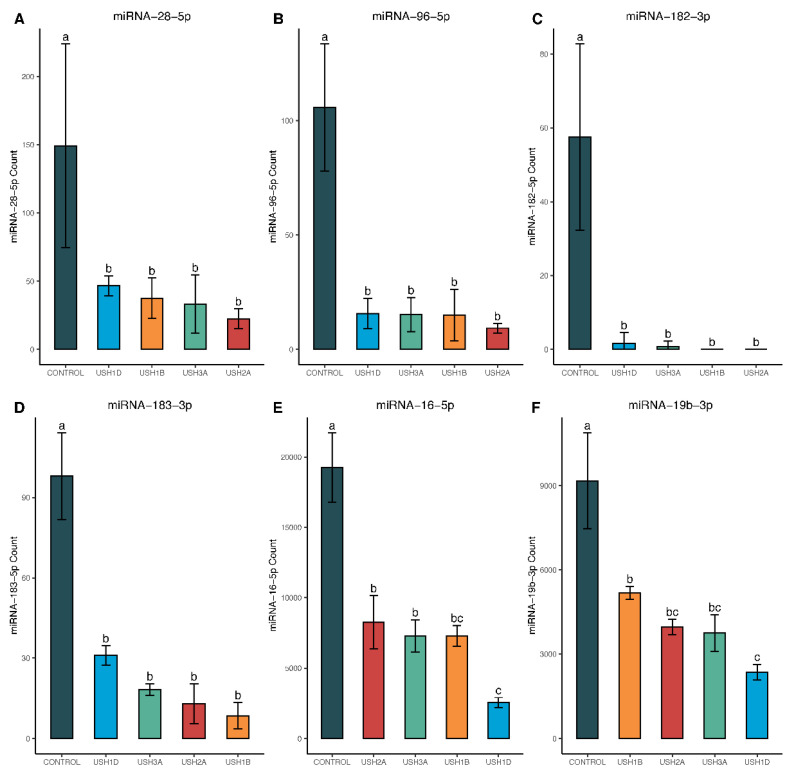
NanoString miRNA counts of 6 downregulated candidate miRNAs in Usher cell lines, compared to controls. Statistical comparisons were made for each miRNA using a one-way ANOVA with Tukey’s adjustment for multiple comparisons, with significance denoted at *p* < 0.05. Genotypes with the same letter above the error bars are statistically the same, while differing letters indicate significant differences from one another (i.e., *p*-values ≤ 0.05). As shown, each panel (**A**–**F**) represents an individual miRNA.

**Figure 5 ijms-25-09993-f005:**
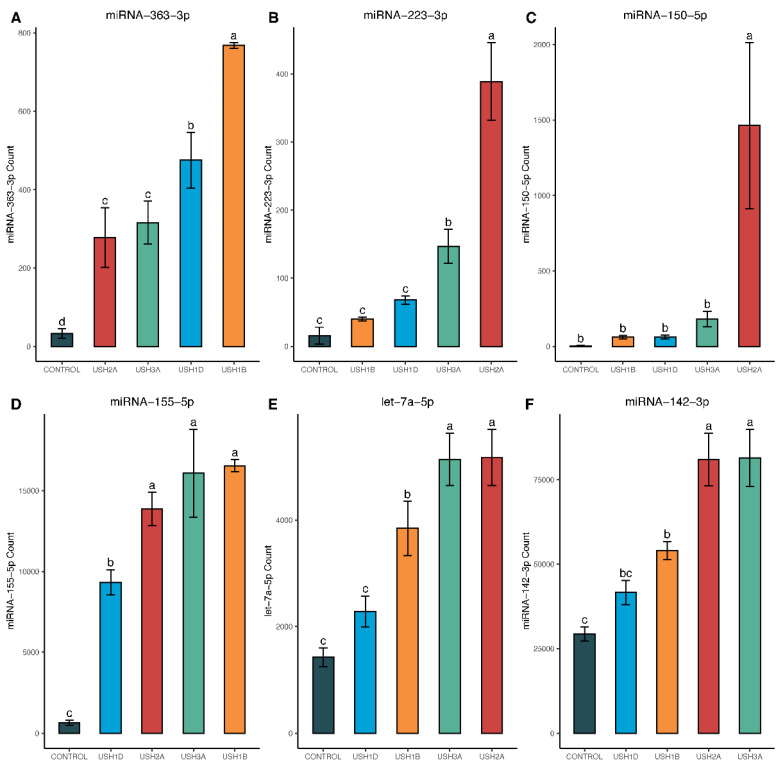
NanoString miRNA counts of 6 upregulated candidate miRNAs in Usher cell lines, compared to controls. Statistical comparisons were made for each miRNA using a one-way ANOVA with Tukey’s adjustment for multiple comparisons, with significance denoted at *p* ≤ 0.05. Genotypes with the same letter above the error bars are statistically the same, while differing letters indicate significant differences from one another (i.e., *p*-values ≤ 0.05). As shown, each panel (**A**–**F**) represents an individual miRNA.

**Figure 6 ijms-25-09993-f006:**
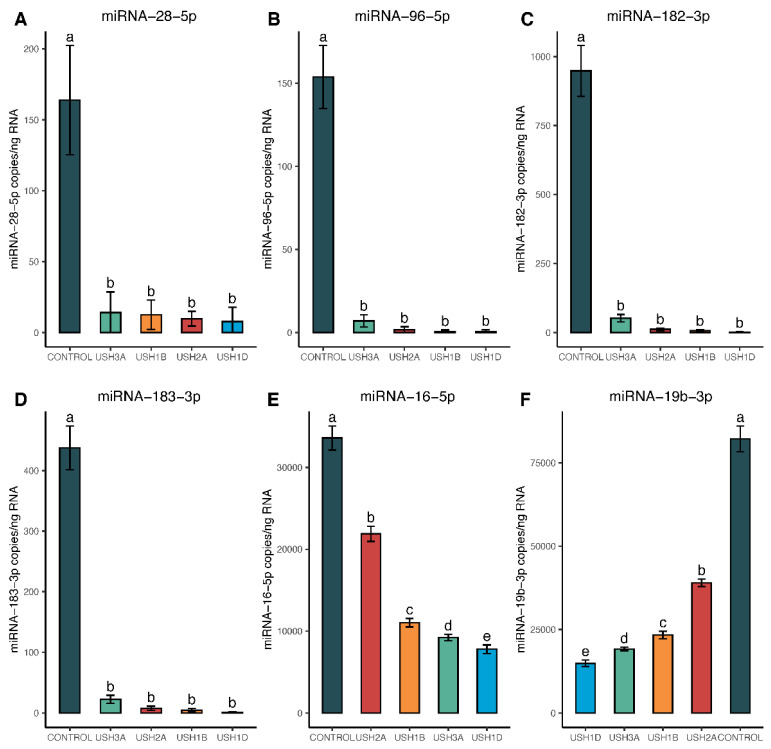
Quantitative analysis of 6 downregulated candidate miRNAs in Usher cell lines, compared to controls using droplet digital PCR technology. Statistical comparisons were made for each miRNA using a one-way ANOVA with Tukey’s adjustment for multiple comparisons, with significance denoted at *p* < 0.05. Genotypes with the same letter above the error bars are statistically the same, while differing letters indicate significant differences from one another (i.e., *p*-values ≤ 0.05). As shown, each panel (**A**–**F**) represents an individual miRNA.

**Figure 7 ijms-25-09993-f007:**
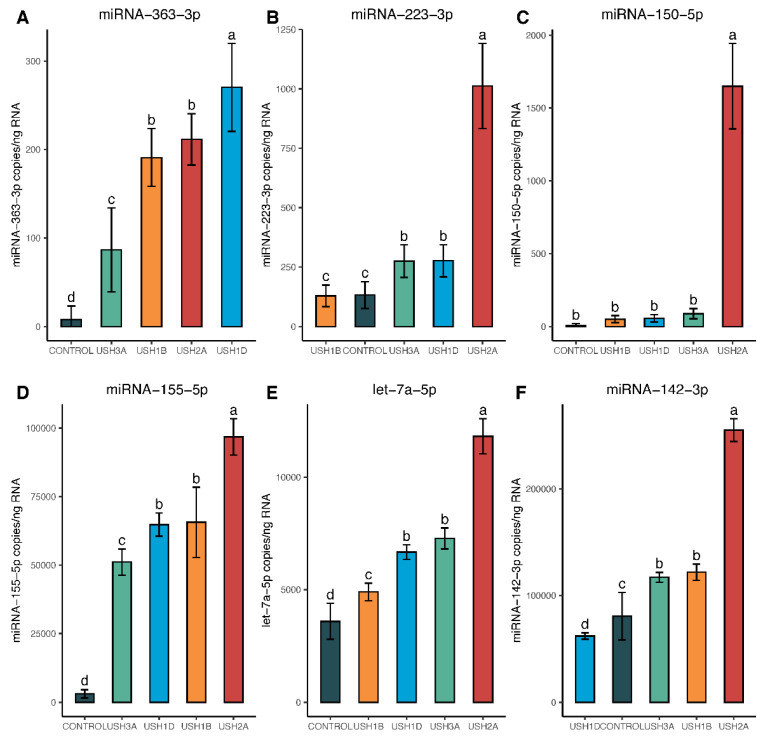
Quantitative analysis of 6 upregulated candidate miRNAs in Usher cell lines, compared to controls using droplet digital PCR technology. Statistical comparisons were made for each miRNA using a one-way ANOVA with Tukey’s adjustment for multiple comparisons, with significance denoted at *p* < 0.05. Genotypes with the same letter above the error bars are statistically the same, while differing letters indicate significant differences from one another (i.e., *p*-values ≤ 0.05). As shown, each panel (**A**–**F**) represents an individual miRNA.

**Table 1 ijms-25-09993-t001:** Exon capture of sequencing variants classified using VarSeq. For each cell line, the phenotype that was clinically diagnosed was verified using NGS targeted exon sequencing where likely pathogenic mutations were documented. Chromosomal locations of the mutations, the reference base call, and the sample mutation are all provided. The American College of Medical Genetics (ACMG) classification criteria, as well as the Human Genome Variant Society (HGVS) variant nomenclature, were followed to describe genes affected, zygosity, predicted inheritance, and known associated conditions for all likely pathogenic mutations.

	Cell Line			
	D3739	D3741	D2880	Coriell (GM09053)
Phenotype	Usher-1D	Usher-1B	Usher-3A	Usher-2A
Chr:Pos	10:71779316	11:77181589	3:150928107	1:216190280 1:215647526
Ref/Alt	G/A	G/T	A/C	AG/- T/-
Genotype	Homozygous	Homozygous	Homozygous	Heterozygous Heterozygous
Classification	Likely Pathogenic	Likely Pathogenic	VUS/Conflicting	Pathogenic Pathogenic
ACMG Criteria	PM2, PS1, PP3	PM2, PP2, PS1, PP3	BS1, PVS1 Strong, PP5	PM2, PVS1, PP5 PM2, PVS1, PP5
HGVS cDot	NM_022124.6:c.5237G>A	NM_000260.4:c.2904G>T	NM_174878.3:c.528T>G	NM_206933.4:c.4338_4339delCT NM_206933.4:c.14787delA
Seq. Ontology	missense	missense	stop gained	frameshift frameshift
Gene Name	*CDH23*	*MYO7A*	*CLRN1*	*USH2USH2A*
Inheritance	Recessive	Recessive	Recessive	Recessive Recessive
Conditions	Usher syndrome type 1D, *CDH23*-Related Disorders, Autosomal recessive nonsyndromic hearing loss 12, Pituitary adenoma 5, Rare genetic deafness, Retinal dystrophy, Childhood onset hearing loss, Usher syndrome	Rare genetic deafness, Autosomal recessive nonsyndromic hearing loss 2, Usher syndrome type 1B	Usher syndrome type 3, Rare genetic deafness, Retinitis pigmentosa 61; Usher syndrome type 3A	Usher syndrome type 2A, *USH2A*-Related Disorders, Retinal dystrophy, Retinitis pigmentosa 39; Usher syndrome type 2A

**Table 2 ijms-25-09993-t002:** Differentially expressed miRNAs from microarray using DESeq2. A summary of all differentially abundant miRNAs expressed by cell lines, shown as clusters—“Common to all” (includes all Usher-type cell lines compared to controls); specifically identified miRNAs separating Usher 1, Usher 2 and Usher 3; and other miRNAs common between Usher 1 and 2, Usher 1 and 3, and Usher 2 and 3.

	Number	Differentially Abundant microRNAs
Common To All	36	hsa-miR-222-3p, hsa-miR-424-5p, hsa-miR-503-5p, hsa-miR-1252-5p, hsa-miR-1246, hsa-miR-182-5p, hsa-miR-1205, hsa-miR-450a-5p, hsa-miR-654-5p, hsa-miR-3934-5p, hsa-miR-221-5p, hsa-miR-34a-5p, hsa-miR-150-5p, hsa-let-7d-5p, hsa-miR-483-3p, hsa-miR-146b-5p, hsa-miR-5010-3p, hsa-miR-591, hsa-miR-155-5p, hsa-miR-363-3p, hsa-let-7i-5p, hsa-miR-194-5p, hsa-miR-28-3p, hsa-miR-10a-5p, hsa-miR-146a-5p, hsa-miR-337-3p, hsa-miR-200c-3p, hsa-miR-96-5p, hsa-miR-5196-3p+hsa-miR-6732-3p, hsa-let-7f-5p, hsa-miR-577, hsa-let-7g-5p, hsa-miR-183-5p, hsa-let-7b-5p, hsa-miR-148a-3p, hsa-miR-98-5p
Usher 1 Only	20	hsa-miR-151a-5p, hsa-miR-129-2-3p, hsa-miR-519d-3p, hsa-miR-4431, hsa-miR-345-5p, hsa-miR-152-3p, hsa-miR-1260b, hsa-miR-3916, hsa-miR-4455,hsa-miR-3195, hsa-miR-195-5p, hsa-miR-1226-3p, hsa-miR-151a-3p, hsa-miR-181b-2-3p, hsa-miR-1287-5p, hsa-miR-324-5p, hsa-miR-484, hsa-miR-16-5p, hsa-miR-331-3p, hsa-miR-26b-5p
Usher 2 Only	14	hsa-miR-513b-5p, hsa-miR-137, hsa-miR-370-3p, hsa-miR-503-3p, hsa-miR-767-5p, hsa-miR-132-3p, hsa-let-7c-5p, hsa-miR-1304-5p, hsa-miR-514a-3p, hsa-let-7a-5p, hsa-miR-342-3p, hsa-miR-181a-5p, hsa-miR-181c-5p, hsa-miR-1193
Usher 3 Only	5	hsa-miR-125a-3p, hsa-miR-494-3p, hsa-miR-125b-5p, hsa-miR-4284, hsa-miR-181a-2-3p
Usher 1 and 2	2	hsa-miR-551b-3p, hsa-miR-299-3p
Usher 1 and 3	5	hsa-miR-371b-5p, hsa-miR-601, hsa-miR-1827, hsa-miR-27a-3p, hsa-miR-582-5p
Usher 2 and 3	10	hsa-miR-574-3p, hsa-miR-223-3p, hsa-miR-1183, hsa-miR-28-5p, hsa-miR-542-5p, hsa-let-7e-5p, hsa-miR-23c, hsa-miR-221-3p, hsa-miR-374b-5p, hsa-miR-23a-3p

**Table 3 ijms-25-09993-t003:** Top 12 differentially abundant miRNAs. A summary table documenting the top 12 differentially expressed miRNAs from the study. Log2 fold change (log2FC) as well as BH-adjusted p-values for each miRNA in each Usher type when compared to the control lines are shown in each row. The final two columns show the average log2FC and standard deviation of all cell lines. Adjusted *p*-values < 0.00001 are noted as zero.

microRNA	Usher 1		Usher 2		Usher 3		Mean log2FC vs. Control	
	log2FC	*p*-Value (BH-adj)	log2FC	*p*-Value (BH-adj)	log2FC	*p*-Value (BH-adj)	Mean log2FC	Standard Dev.
hsa-miR-150-5p	4.8901	0	9.7786	0	6.5893	0	7.086	2.4818
hsa-miR-155-5p	4.15	0	4.6412	0	4.6898	0	4.4936	0.2986
hsa-miR-363-3p	4.1282	0	3.3498	0	3.3819	0	3.62	0.4404
hsa-miR-223-3p	1.6878	0.0002	4.804	0	3.2343	0	3.242	1.5581
hsa-let-7a-5p	0.9413	0.0095	2.0768	0	1.9006	0	1.6396	0.6111
hsa-miR-142-3p	0.5645	0.0259	1.6779	0	1.5316	0	1.258	0.6051
hsa-miR-19b-3p	−1.4389	0.0001	−0.9554	0.0293	−1.1786	0.0041	−1.191	0.2419
hsa-miR-28-5p	−1.9993	0	−2.5731	0	−2.0932	0	−2.2219	0.3078
hsa-miR-16-5p	−2.1845	0	−1.0125	0.0487	−1.3422	0.0044	−1.513	0.6044
hsa-miR-183-5p	−2.2771	0	−2.6795	0	−2.3467	0	−2.4344	0.2151
hsa-miR-96-5p	−2.9386	0	−3.2701	0	−2.7408	0	−2.9832	0.2674
hsa-miR-182-5p	−6.2691	0.0024	−8.0376	0.0005	−6.1735	0.0073	−6.8267	1.0498

## Data Availability

The raw data supporting the conclusions of this article are available from the authors without any reservation. Raw data and code for the statistical analysis of microarray and ddPCR data can be obtained from https://github.com/westom21/Usher_miRNA, 1 August 2024. Targeted exon raw sequencing data can be accessed from NCBIs SRA by searching the project number PRJNA1063720.

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
