# Peer review of "Genotype Characterization and MiRNA Expression Profiling in Usher Syndrome Cell Lines"

_ijms, 2024, doi:10.3390/ijms25189993_

Round 1
Reviewer 1 Report
Comments and Suggestions for Authors Usher syndrome is a disorder that affects the visual, auditory and vestibular systems. In this pathology, the development of the cilliad cells of the ear and the stibular system and the cells of the retina are altered. In addition to clinical tests to assess hearing, vision and balance, genetic tests have been developed for genes that have been associated with Usher syndrome such as MYO7A, USH1C, CDH23, PCDH23, USH1E, USH1G, USH1K, CIB2/DFNB48, USH2A genes, GPR98, WHRN, CLRN-1, HARS and PDZD7. However, it is necessary to continue development of new biomarkers that allow early diagnosis and management of the disease. The authors have studied the miRNA expression profiles in different types of Usher cell lines and derived controls, from which they obtained a list of potential miRNAs that are validated by ddPCR. Since it is a hereditary disease, the genetic profile of the affected cells in the body should be the same as that of the lymphocytes from which the cell models are generated. And this should not be affected by age. However, epigenetic markers, such as miRNAs, depend on the cell type and age, and more so taking into account that the disease develops at different times, so it should be checked that at least the age of the control donors is matched with patients. It should also be checked whether EBV changes the miRNA profile. The authors indicate that they have identified miRNAs specific to USH1, USH2 and USH3. But if what I said before is confirmed, they would really be looking at miRNAs whose expression depends on the mutated genes CDH23, MYO7A, CLRN1 and USH2A. To affirm that they are miRNAs of the disease, shouldn't the same be done for the rest of the known mutations? An alternative to using lymphocytes, which do not reflect the real environment of the disease, could be to use retinal and ciliated cells, generate targeted mutations against the known genes for USH and evaluate the levels of miRNAs. Given the difficulty of my proposal, could you analyze the profile of circulating miRNAs in patients and carriers of the disease to confirm the results? They make microarrays, after validating them by ddPCR. They should analyze bioinformatics to see which pathways are alteredAuthor Response
Reviewer 1
Usher syndrome is a disorder that affects the visual, auditory and vestibular systems. In this pathology, the development of the cilliad cells of the ear and the stibular system and the cells of the retina are altered. In addition to clinical tests to assess hearing, vision and balance, genetic tests have been developed for genes that have been associated with Usher syndrome such as MYO7A, USH1C, CDH23, PCDH23, USH1E, USH1G, USH1K, CIB2/DFNB48, USH2A genes, GPR98, WHRN, CLRN-1, HARS and PDZD7. However, it is necessary to continue development of new biomarkers that allow early diagnosis and management of the disease. The authors have studied the miRNA expression profiles in different types of Usher cell lines and derived controls, from which they obtained a list of potential miRNAs that are validated by ddPCR. Since it is a hereditary disease, the genetic profile of the affected cells in the body should be the same as that of the lymphocytes from which the cell models are generated. And this should not be affected by age.
Dear Sir/Madam
We, the authors of this manuscript would like to take this opportunity to thank you for taking your valuable time and your expertise to review our manuscript. Please see below for our responses to the issues raised by you.
[Comment 1] However, epigenetic markers, such as miRNAs, depend on the cell type and age, and more so taking into account that the disease develops at different times, so it should be checked that at least the age of the control donors is matched with patients.
[Response 1]
Thank you raising this important question. You are 100% right when you say miRNA expression profiles depend on cell types, age and lot of other factors. It is true that miRNA expression profiles in patient’s affected tissues (inner ear tissues, retina and vestibular cells) may be different from miRNA expression profiles we observed in lymphocytes obtained from usher patients. Our goal was not to study the miRNA expression profiles in affected tissues to study how those profiles are related to disease conditions but to find changes in miRNA profiles in lymphocytes due to Usher causing mutations, so those dysregulated miRNAs can be used as biomarkers to diagnose/screen usher syndrome. Please read the last part of our discussion section where we have reported limitations of our study. From line 315 to 326.
In our current study we used 4 usher cell lines. Two usher type 1 cell lines, one usher type 2 cell line and one usher type 3 cell line. The usher type 2 cell line we used was purchased from Corial Institute and the age of patient was nine years at the time of lymphocyte collection. For the other 3 cell lines patient’s age at the time of lymphocyte collection is not available. Therefore, we are not able to produce age matched controls without that vital information.
In addition, we would like to emphasize that we are interested in developing a test which is applicable for Usher diagnosis regardless of the age of the patient. For example, we see that miRNA 183-182-96 family is universally significantly downregulated in all usher cell lines compared to our control. Regardless of the age of the patient at the collection of their B-lymphocytes, we see the same expression trend.
[Comment 2] It should also be checked whether EBV changes the miRNA profile.
[Response 2]
Thank you for raising this question. It has been reported that immortalizing cells using EBV changes miRNA expression profiles in cells. It is fairly well documented that miRNA expression profiles change in host B-lymphocytes upon exposure to EBV, especially miRNA 155, which is typically elevated upon EBV infection and is thought to play a role in immune system evasion and cell proliferation (PMID: 18753206, PMID: 32194570, PMID: 23246696, PMID: 22496226). Interestingly, miRNA 155 is a member of our panel, yet expression in the control cell line (NBT) remains highly downregulated. So despite what is reported in the literature, even within our immortalized cell types there is still an expression discrepancy between Usher and control lines, even in miRNAs known to be changed due to EBV infection. Since we knew that EBV cause changes in miRNA expression profiles in lymphocytes, we used normal immortalized cells as our control. Both Normal hearing control samples as well as Usher patient-derived lymphocytes were purified from blood and immortalized following standard EBV-transformation protocol. Therefore, we suspect that all miRNAs shown to express differentially in Usher cells compared to control, are due to inherent pathologies or gene mutations related to Usher syndrome.
References:
Lu F, Weidmer A, Liu CG, Volinia S, Croce CM, Lieberman PM. Epstein-Barr virus-induced miR-155 attenuates NF-kappaB signaling and stabilizes latent virus persistence. J Virol. 2008 Nov;82(21):10436-43. doi: 10.1128/JVI.00752-08. Epub 2008 Aug 27. PMID: 18753206; PMCID: PMC2573162.
Forte E, Salinas RE, Chang C, Zhou T, Linnstaedt SD, Gottwein E, Jacobs C, Jima D, Li QJ, Dave SS, Luftig MA. The Epstein-Barr virus (EBV)-induced tumor suppressor microRNA MiR-34a is growth promoting in EBV-infected B cells. J Virol. 2012 Jun;86(12):6889-98. doi: 10.1128/JVI.07056-11. Epub 2012 Apr 11. PMID: 22496226; PMCID: PMC3393554.
Elton TS, Selemon H, Elton SM, Parinandi NL. Regulation of the MIR155 host gene in physiological and pathological processes. Gene. 2013 Dec 10;532(1):1-12. doi: 10.1016/j.gene.2012.12.009. Epub 2012 Dec 14. PMID: 23246696.
Iizasa H, Kim H, Kartika AV, Kanehiro Y, Yoshiyama H. Role of Viral and Host microRNAs in Immune Regulation of Epstein-Barr Virus-Associated Diseases. Front Immunol. 2020 Mar 3;11:367. doi: 10.3389/fimmu.2020.00367. Erratum in: Front Immunol. 2020 Apr 03;11:498. doi: 10.3389/fimmu.2020.00498. PMID: 32194570; PMCID: PMC7062708.
[Comment 3] The authors indicate that they have identified miRNAs specific to USH1, USH2 and USH3. But if what I said before is confirmed, they would really be looking at miRNAs whose expression depends on the mutated genes CDH23, MYO7A, CLRN1 and USH2A. To affirm that they are miRNAs of the disease, shouldn't the same be done for the rest of the known mutations?
[Response 3]
Reviewer 1 raises a valid concern that miRNA expression profiles will be different based on the genotype of the patient. Here we classify four genotypes, and the argument is that there are many other genes where mutations would lead to Usher syndrome. What does the miRNA expression look like in the other Usher genotypes? This is a valid question and warrants further investigation. However, to do so would involve finding patients or cell lines with mutations in each of the known associated genes associated with Usher syndrome, and performing miRNA profiling on each specific genotype. Reviewer 1 aptly provided a list of 14 genes linked to Usher. As stated in the introduction of the manuscript, Usher syndrome is a rare genetic disorder, affecting ~4 in 100,000 individuals and collecting patient samples is a difficult endeavor, hence our use of cell lines. For our lab to acquire 10 more genotypes, perform miRNA microarray and ddPCR with replicates would be a significant financial endeavor, and might lie outside the capability of the lab.
Besides, I would like to point to the heatmap of miRNA expression (Figure 3). One of the main observations from this study is that there appears to be generalizations we can make about the miRNAs selected. First, looking at cluster A in figure three, there are 6 miRNAs (28-5p, 183-5p, 182-5p, 96-5p, 16-5p, and 19b-3p) which are significantly downregulated in all Usher genotypes. Thus, we might be able to use this set of six downregulated miRNAs to diagnose a potential Usher patient from a blood draw at an early age. If we examine block B in figure 3, miRNAs 155-5p, 142-3p, and let-7a-5p all are significantly upregulated in all Usher patients compared to our control. This is to say that there are generalized miRNA expressions that are consistent in multiple genotypes which can be used as broad biomarkers. Conversely, as reviewer 1 points out, there are genotype specific observations as well. For example, in figure 3, miRNA 150-5p would be an indicator of the USH2A genotype, while miRNA 363-3p serves as a better indicator of the Usher type 1 genotypes (MYO7A and CDH23). So, while it is true that each genotype will have varying miRNA profiles as a whole, there are generalities that can be derived from the data as well which are described in the discussion lines 215-304. In our future studies we hope to characterize other usher causing mutations and their associated miRNA expression patterns. We have discussed this in our discussion section. (Please refer lines 315 to 326: as we acknowledge the limitations of our current study).
[Comment 4] An alternative to using lymphocytes, which do not reflect the real environment of the disease, could be to use retinal and ciliated cells, generate targeted mutations against the known genes for USH and evaluate the levels of miRNAs.
[Response 4]
Thank you for this insight. We agree that lymphocytes may not reflect the real environment where the disease progresses. And future studies involving alternate cell culture models (retina/ciliated cells) certainly would help evaluate the levels of miRNA expression observed in this Usher cell line-based study. However, defining precise neurosensory/miRNA targets in inner ear remains a major challenge, studies have mostly relied on animal/cell-culture models due to inaccessibility of the human inner ear. Blood (circulating plasma/lymphocytes) would be a less invasive sampling ideal for biomarkers research and Usher diagnosis.
[Comment 5] Given the difficulty of my proposal, could you analyze the profile of circulating miRNAs in patients and carriers of the disease to confirm the results?
[Response 5]
This is a great suggestion. However, this is beyond the scope of current investigation and outside of the capabilities of the lab at present. We do not have a patient population to draw from at the moment.
[Comment 6] They make microarrays, after validating them by ddPCR. They should analyze bioinformatics to see which pathways are altered.
[Response 6]

Above I have included a pathway analysis using gene ontologies based on molecular function using miRPathDB version 2.0 against our 12 miRNA candidates. The miRNAs in our panel are implemented in quite a large number of pathways. MiRNA 142 and 363 are active in only a few pathways, while miRNA’s 155 and 16 have targets in many different pathways. The figure below is the same analysis using KEGG ontologies.
Here we also see a diverse array of possible pathways our 12 miRNAs could be active in. However we would like to make two assertions. First, as reviewer 1 has correctly pointed out, these samples are B-lymphocyte samples and are probably not the best tissue sample to make inference about the effect miRNA expression may or may not be having on the inner ear or retina. Thus, the pathway analysis provided here would not be very applicable for disease causation. We would also like to emphasize here that we are not necessarily aimed at the function of the miRNAs we identified and their gene targets, but rather whether the miRNAs in this study can consistently identify patients with Usher syndrome. We did/do not make any claims that the 12 miRNAs in this study are responsible for the development of Usher syndrome, we are simply observing elevated levels in B-lymphocytes and would like to leverage these miRNAs to be able to identify User syndrome patients.
Reviewer 2 Report
Comments and Suggestions for Authors
Wesley Tom et al. explored novel microRNA biomarkers by extracting lymphocytes from patients and transforming them into stable cell lines. They then performed DNA sequencing to identify gene variants and conducted miRNA microarray analysis to distinguish differential microRNA expression between USH and control cell lines. After reading through the manuscript, my suggestions are:
1. The labels in figure1 are too small compared to the Veen plot.
2. The author aimed to identify markers for the early detection of Usher syndrome. Are the USH patients in this study in the early stages of the disease?
3. Are there any relationships between USH-related gene variants and miRNA?
4. What is the miRNA difference between the Usher-1D and Usher-1B cell lines? Are these differences related to their genotypes?
5. The author identified common miRNAs for USH, it will be interesting and informative to infer the regulated targets by these miRNAs.
Author Response
Reviewer 2
Wesley Tom et al. explored novel microRNA biomarkers by extracting lymphocytes from patients and transforming them into stable cell lines. They then performed DNA sequencing to identify gene variants and conducted miRNA microarray analysis to distinguish differential microRNA expression between USH and control cell lines. After reading through the manuscript, my suggestions are:
Dear Sir/Madam
We, the authors of this manuscript would like to take this opportunity to thank you for taking your valuable time and your expertise to review our manuscript. Please see below for our responses to the issues raised by you.
[Comment 1] The labels in figure1 are too small compared to the Veen plot.
[Response 1]
Thank you for pointing out this issue. In the revised version we have done the correction you requested.
[Comment 2] The author aimed to identify markers for the early detection of Usher syndrome. Are the USH patients in this study in the early stages of the disease?
[Response 2]
Three cell lines (CDH23, MYO7A, CLRN1)were developed at the Boys Town Nation Research Hospital. Blood samples were drawn just after clinically diagnosing the patients. Regarding these three cell lines the answer to your question is yes. USH2A cell line was purchased from Corial Institute and according to their web site the age of the patient at the time of blood draw was 9 years.
[Comment 3] Are there any relationships between USH-related gene variants and miRNA?
[Response 3]
In this study we investigated the relationship between some usher gene variants and miRNA in lymphocytes. We have found some dysregulated miRNAs in usher lymphocytes compared to control lymphocytes, see lines 251-304 a large fraction of the manuscript attempts to discuss how the 12 miRNAs. To the best of our knowledge, the 12 miRNAs in the study are not known to act upon the genes where we find mutations causing the various Usher genotypes. We did this by mining known miRNA databases (miRDB, miRbase) for our microRNAS, and their potential associations with MYO7A, USH2A, CDH23, and CLRN-1genes and we found no significant binding sites or associations.
[Comment 4] What is the miRNA difference between the Usher-1D and Usher-1B cell lines? Are these differences related to their genotypes?
[Response 4]
Please refer to the last part of our discussion section (lines 318 to 323) which reads like:
“Second, it's worth noting that miRNA expression patterns may likely vary depending on the patient's genotype. In our study, we observed significant differences in miRNA profiles within both USH1B and USH1D genotypes, despite both falling under the USH1 classification. This divergence is attributed to distinct sequence variants found in different genes, contributing to the observed phenotypic differences”. Usher I subtypes (1B and 1D) are also distinguishable by miRNA due to unique expression levels of specific miRNAs: possibly related to Usher-I genotypes: *miR-let-7a-5p upregulated in all Usher (including Ush-1B), but remained unchanged in Ush-1D; *miR-223-3p upregulated in all Usher (including Ush-1D), except Ush-1B.
Usher 1 subtypes (Usher-1D and Usher-1B) are indeed classified by the genes which possess the dominant pathogenic mutation, although there are digenic instances reported in Usher patients. Usher-1B (MYO7A mutations) make up 53-70% of Usher 1 pathogenic variants, while Usher-1D (CDH23) mutations account for between 10-20% of pathogenic variants in Usher 1 patients.
Reference:
Koenekoop RK, Arriaga MA, Trzupek KM, et al. Usher Syndrome Type I. 1999 Dec 10 [Updated 2020 Oct 8]. In: Adam MP, Feldman J, Mirzaa GM, et al., editors. GeneReviews® [Internet]. Seattle (WA): University of Washington, Seattle; 1993-2024. Available from: https://www.ncbi.nlm.nih.gov/books/NBK1265/
[Comment 5] The author identified common miRNAs for USH, it will be interesting and informative to infer the regulated targets by these miRNAs.
[Response 5]

Above I have included a pathway analysis using gene ontologies based on molecular function using miRPathDB version 2.0 against our 12 miRNA candidates. The miRNAs in our panel are implemented in quite a large number of pathways. MiRNA 142 and 363 are active in only a few pathways, while miRNA’s 155 and 16 have targets in many different pathways. The figure below is the same analysis using KEGG ontologies.

Here we also see a diverse array of possible pathways our 12 miRNAs could be active in. However, we would like to make two assertions. First, as reviewer 1 has correctly pointed out, these samples are B-lymphocyte samples and are probably not the best tissue sample to make inference about the effect miRNA expression may or may not be having on the inner ear or retina. Thus, the pathway analysis provided here would not be very applicable for disease causation. We would also like to emphasize here that we are not necessarily aimed at the function of the miRNAs we identified and their gene targets, but rather whether the miRNAs in this study can consistently identify patients with Usher syndrome. We did/do not make any claims that the 12 miRNAs in this study are responsible for the development of Usher syndrome, we are simply observing elevated levels in B-lymphocytes and would like to leverage these miRNAs to be able to identify User syndrome patients.
Round 2
Reviewer 1 Report
Comments and Suggestions for Authors
In my opinion, although the work focuses on an area of ​​interest, the authors should perform more experiments or completely refocus the writing of the work. If it is a biomarker study, they should recruit a large number of patients and look for circulating biomarkers or lymphocytes, ... and the work carried out on the transformation of lymphocytes with typical genotypes in USH would be to select or demonstrate that the identified miRNAs could be related to the disease. But this would not be strictly necessary in a study of circulating biomarkers.
However, as the work currently stands it seems more like a mechanistic study, in which much remains to be done.
Author Response
“Genotype Characterization and MiRNA Expression Profiling in Usher Syndrome Cell Lines”
Rev>1
More experiments – May help better insights of mechanisms/pathways in Usher Syndrome, but this would certainly remain as future directions.
Refocus writing work – The revised MS is more focused and illustrates miRNA differential expression patterns in Lymphocyte cell lines derived from Usher patients and healthy control.
Biomarker study / needs large Pt population - The title has been revised as per editors’ suggestion and now primarily focuses on miRNA profiling comparing Usher and Control cell lines, with no claim on biomarker discovery.
Identified miRNA could be related to disease – The study describes top 12 miRNAs with differential expression patterns that may distinguish the Usher Syndrome (including subtypes 1B/1D), however, aim of the study is not to describe any direct link to Usher disease. As suggested, this will be a subject of future studies that target larger Pt populations.
Not strictly necessary as the study focus is on circulating biomarkers – We appreciate the reviewers understanding about limitations of our study, and that it only focuses on miRNA expression patterns in circulating Lymphocytes not confined locally to any disease sites (inner ear).
We acknowledge that further research linking miRNAs to Usher disease subtypes would help mechanistic insights of the disease – in which much remains to be done!